# Molecular Systematics and Taxonomic Analyses of Three New Wood-Inhabiting Fungi of *Hyphoderma* (Hyphodermataceae, Basidiomycota)

**DOI:** 10.3390/jof9111044

**Published:** 2023-10-24

**Authors:** Yang Yang, Qianquan Jiang, Qi Li, Jiawei Yang, Li Cha, Lijun Cheng, Shunqiang Yang, Changlin Zhao, Hongmin Zhou

**Affiliations:** 1College of Biodiversity Conservation, Southwest Forestry University, Kunming 650224, China; fungiyoung@163.com (Y.Y.); fungiqianquanjiang@163.com (Q.J.); fungiqili@163.com (Q.L.); 2Yunnan Key Laboratory of Gastrodia and Fungal Symbiotic Biology, Zhaotong University, Zhaotong 657000, China; 3Key Laboratory for Forest Resources Conservation and Utilization in the Southwest Mountains of China, Ministry of Education, Southwest Forestry University, Kunming 650224, China; 4Office of Management and Protection, Green Peacock Provincial Nature Reserve, Dali 671000, China

**Keywords:** biodiversity, ecology, molecular systematics, multi-gene, wood-inhabiting fungi, Yunnan Province

## Abstract

In this present study, three new wood-inhabiting fungal taxa, *Hyphoderma niveomarginatum*, *H. sordidum* and *H. weishanense,* are proposed. *Hyphoderma niveomarginatum* is characterized by the ceraceous basidiomata having a smooth, cracking hymenial surface and the presence of the moniliform cystidia and ellipsoid basidiospores (7–9 × 3.5–5 µm). *Hyphoderma sordidum* is characterized by its resupinate basidiomata with a smooth hymenial surface with the fimbriate margin, the presence of the tubular cystidia and ellipsoid basidiospores (3–4.5 × 2–3 µm). *Hyphoderma weishanense* differs in its membranous basidiomata with a slightly buff to buff hymenial surface and the presence of broadly ellipsoid basidiospores (4.5–8.5 × 4–7 µm). Sequences of ITS+nLSU+mt-SSU+RPB1+RPB2 genes were used for the phylogenetic analyses using three methods. The ITS+nLSU+mt-SSU+RPB1+RPB2 analysis of the genus *Hyphoderma* indicated that the 3 new species of *Hyphoderma* were nested into genus *Hyphoderma*, in which *H. niveomarginatum* formed a single group and then grouped with *H. membranaceum* and *H. sinense*; *H. sordidum* was a sister to *H. nudicephalum*; and *H. weishanense* closely grouped with *H. crystallinum*.

## 1. Introduction

Fungi are eukaryotic microorganisms that play fundamental ecological roles as decomposers and mutualists of plants and animals, in which they drive carbon cycling in the forest ecosystem, mediate the mineral nutrition of plants and alleviate carbon limitations of other soil organisms [1]. Wood-inhabiting fungi are an ecologically important branch of the tree of life, inferred from their distinct and diverse characteristics [2]. Taxa from the family Hyphodermataceae are continuously reported, with the frequent inclusion of data from DNA sequences and by employing both fresh material and cultures, and mycologists re-collect historic taxa and their types to accomplish taxonomy and molecular systematics in this family Hyphodermataceae [3,4].

The genus *Hyphoderma* Wallr. (1833: 576) belongs to the family Hyphodermataceae (Polyporales, Basidiomycota), typified by *H. setigerum* (Fr.) Donk. (1957: 15), and represents one of the important genera among wood-inhabiting fungi [4]. The genus is characterized by resupinate to effuse-reflexed basidiomata with ceraceous consistency, and smooth to tuberculate, grandinioid or odontioid hymenial surfaces, a monomitic hyphal system (rarely dimitic) with clamp connections on generative hyphae, the presence or not of cystidia, suburniform to subcylindrical and cylindrical basidia, and ellipsoid to subglobose, smooth, thin-walled basidiospores [5]. Based on the Index Fungorum (www.indexfungorum.org; accessed on 3 September 2023), the genus *Hyphoderma* has 206 specific and registered names. Currently, one hundred and thirteen species have been accepted worldwide [6,7,8,9,10,11,12,13].

This pioneering research for the phylogenetic analysis process of the genus *Hyphoderma* was just the prelude to the molecular systematics period [9,10,11,12,13,14]. The phylogenetic research revealed that all *Hyphoderma* taxa clustered into the different groups in phylogenetic trees at the class level based on the molecular phylogenetic methods, in which the result indicated that *H. praetermissum* (P. Karst.) J. Erikss. & Å. Strid and *Resinicium bicolor* (Alb. & Schwein.) Parmasto were grouped together, while the other *Hyphoderma* species, *Hypochnicium* J. Erikss, and several other species formed a separate branch [14]. The phylogeny of *Hyphoderma* showed that two species *H. obtusum* J. Erikss. and *H. setigerum* nested into the family Meruliaceae Rea and formed a sister taxon to *Hypochnicium polonense* (Bres.) Å. Strid [15]. The phylogenetical relationships among the closely related taxa in *Hyphoderma* were determined and a new species was proposed, *H. macaronesicum* Tellería, M. Dueñas, Beltrán-Tej., Rodr-Armas & M.P. Martín [16]. The research comprising the representative sequences of the *H. setigerum* complex showed that *H. pinicola* Yurch. & Sheng H. Wu represented a fifth species in this complex of this genus *Hyphoderma* [17]. The research of the family-level classification of the order Polyporales indicated that four *Hyphoderma* species grouped into the residual polyporoid clade, belonging to the family Hyphodermataceae, in which they grouped with three related genera in the family Meripilaceae as *Meripilus* P. Karst., *Physisporinus* P. Karst. and *Rigidoporus* Murrill [18].

During the investigations of the wood-inhabiting fungi, we collected three new *Hyphoderma* taxa from Yunnan Province, China, that could not be assigned to any described species of the order Polyporales. We present the morphological characteristics and multi-gene phylogenetic analyses with nLSU, ITS, mt-SSU, RPB2 and RPB1 that support the three species in the genus *Hyphoderma*.

## 2. Materials and Methods

### 2.1. Sample Collection and Herbarium Specimen Preparation

Fresh fruiting bodies of fungi growing on angiosperm branches were collected from the Lincang, Qujing and Dali of Yunnan Province, China. The samples were photographed in situ and fresh macroscopic details were recorded. Photographs were recorded using a Jianeng 80D camera (Tokyo, Japan). All of the photos were stacked and merged using Helicon Focus Pro 7.7.5 software. Specimens were dried in an electric food dehydrator at 40 °C, and then sealed and stored in an envelope bag and deposited in the herbarium of the Southwest Forestry University (SWFC), Kunming, Yunnan Province, China.

### 2.2. Morphology

Macromorphological descriptions were based on field notes and photos captured in the field and lab. Color terminology followed Petersen [19]. Micromorphological data were obtained from the dried specimens following observation under a light microscope [20]. The following abbreviations were used: KOH = 5% potassium hydroxide water solution, CB = cotton clue, CB– = acyanophilous, IKI = Melzer’s reagent, IKI– = both inamyloid and indextrinoid, L = mean spore length (arithmetic average for all spores), W = mean spore width (arithmetic average for all spores), Q = variation in the L/W ratios between the specimens studied and n = a/b (number of spores (a) measured from a given number (b) of specimens).

### 2.3. DNA Extraction and Sequencing

The EZNA HP Fungal DNA Kit (Omega Biotechnologies Co., Ltd., Kunming, China) was used to extract DNA from the dried specimens. The ITS region was amplified with the primer pair ITS5/ITS4 [21], the nLSU region with the primer pair LR0R/LR7 [22], the mt-SSU region with the primer pair MS1/MS2 [21], the RPB1 region with the primer pair RPB1-Af/RPB1-Cf [23] and the RPB2 region with the primer pair bRPB2-6F/bRPB2-7.1R [24]. The PCR procedure for ITS, nLSU, mt-SSU, RPB1 and RPB2 followed a previous study [22]. All of the newly generated sequences were deposited in GenBank (Table 1).

### 2.4. Phylogenetic Analyses

The sequences were aligned in MAFFT version 7 using the G-INS-i strategy [28]. The alignment was manually adjusted using AliView version 1.27 [29]. The sequence alignments were deposited in TreeBase (ID 30751; (accessed on 8 September 2023)). *Diplomitoporus crustulinus* (Bres.) Domański were assigned as an outgroup to root trees following a previous study analysis [13].

Maximum parsimony (MP), maximum likelihood (ML) and Bayesian Inference (BI) analyses were applied to the three combined datasets. The phylogenetic analysis method was adopted by Zhao and Wu [30]. MP analysis was performed in PAUP* version 4.0b10 [31]. All of the characteristics were equally weighted and gaps were treated as missing data. Trees were inferred using the heuristic search option with TBR branch swapping and 1000 random sequence additions. Max-trees were set to 5000, branches of zero length were collapsed and all most-parsimonious trees were saved. Clade robustness was assessed using bootstrap (BT) analysis with 1000 replicates [32]. Descriptive tree statistics tree length (TL), the consistency index (CI), the retention index (RI), the rescaled consistency index (RC) and the homoplasy index (HI) were calculated for each most-parsimonious tree generated. ML was inferred using RAxML-HPC2 through the Cipres Science Gateway (www.phylo.org (accessed on 13 September 2023)) [33]. Branch support (BS) for ML analysis was determined using 1000 bootstrap replicates and evaluated under the gamma model.

MrModeltest 2.3 [34] was used to determine the best-fit evolution model for each dataset for Bayesian inference (BI), which was performed using MrBayes 3.2.7a with a GTR+I+G model of DNA substitution and a gamma distribution rate variation across sites [35]. Four Markov chains were run twice from a random starting tree, over 10 million generations of the dataset (Figure 1), and the tree was sampled every 1000 generations. The first one fourth of all generations were discarded as burn-in. The majority rule consensus tree of all remaining trees was calculated. Branches were considered as significantly supported if they received a maximum likelihood bootstrap value (BS) >70%, maximum parsimony bootstrap value (BT) >70% or Bayesian posterior probabilities (BPPs) >0.95.

## 3. Results

### 3.1. Molecular Phylogeny

The dataset based on ITS+nLSU+mt-SSU+RPB1+RPB2 (Figure 1) comprises sequences from 60 fungal specimens representing 39 species. The alignment length of this dataset is 5675 characters, of which 2964 characters are constant, 1380 characters are variable with no information and 1331 characters have no information. Maximum parsimony analysis yielded three equally parsimonious trees (TL = 6130, CI = 0.5863, HI = 0.4137, RI = 0.6036, RC = 0.3539). Bayesian analysis and ML analysis resulted in a similar topology as MP analysis with an average standard deviation of split frequencies of 0.025189 (BI), and the effective sample size (ESS) across the two runs is double the average ESS (avg ESS) = 2823.

The phylogram based on the ITS+nLSU+mt-SSU+RPB1+RPB2 rDNA gene regions (Figure 1) indicated that three new species grouped into genus *Hyphoderma*, in which *H. niveomarginatum* grouped with two taxa, *H. membranaceum* C.L. Zhao & Q.X. Guan and *H. sinense* C.L. Zhao & Q.X. Guan, and then closely grouped with *H. transiens* (Bres.) Parmasto, *H. amoenum* (Burt) Donk and *H. fissuratum* C.L. Zhao & X. Ma. *Hyphoderma sordidum* clustered with *H. nudicephalum* Gilb. & M. Blackw. *Hyphoderma weishanense* grouped with *H. crystallinum* C.L. Zhao & Q.X. Guan, and then clustered with *H. variolosum* Boidin, Lanq. & Gilles, *H. marginatum* Z.Y. Duan & C.L. Zhao, *H. medioburiense* (Burt) Donk, *H. assimile* (H.S. Jacks. & Dearden) Donk and *H. subsetigerum* Sheng H. Wu.

### 3.2. Taxonomy

***Hyphoderma niveomarginatum*** Y. Yang & C.L. Zhao, sp. nov. Figure 2 and Figure 3.

MycoBank no.: 849948.

**Holotype—**China, Yunnan Province, Lincang, Yun County, Dumu Village. GPS coordinates: 24°23′ N, 101°9′ E; altitude: 1960 m asl. On fallen unidentified angiosperm branch, leg. C.L. Zhao, 20 October 2022, CLZhao 25078 (SWFC).

**Etymology—*niveomarginatum*** (Lat.): referring to the white margin of basidiomata surface.

**Basidiomata—**Annual, resupinate, adnate, ceraceous, odorless when fresh, and up to 4 cm long, 2 cm wide and 50–200 µm thick. Hymenial surface smooth, pale yellowish when fresh, cream on drying, cracking. Sterile margin white to cream, up to 2 mm wide.

**Hyphal system—**Monomitic; generative hyphae with clamp connections; colorless, thin-walled, frequently branched, interwoven, 2–4 µm in diameter; IKI–, CB–, tissues unchanged in KOH.

**Hymenium—**Cystidia moniliform, with a variable number of constrictions, colorless, thin-walled, 29–55.5 × 5–7 µm; basidia clavate, slightly constricted in the median to somewhat sinuous, often with oil droplets, with four sterigmata and a basal clamp connection, 22–24 × 7–8 µm; basidioles in shape similar to basidia, but slightly smaller.

**Spores—**Basidiospores ellipsoid, colorless, thin-walled, smooth, with irregular oil droplets inside, IKI–, CB–, (6–)7–9(–10) × (3–)3.5–5(–5.5) µm, L = 8.01 µm, W = 4.24 µm, Q = 1.90 (n = 30/1).

**Notes—***Hyphoderma litschaueri* (Burt) J. Erikss. & Å. Strid, *H. malenconii* (Manjón & G. Moreno) Manjón, G. Moreno & Hjortstam, *H. membranaceum*, *H. moniliforme* (P.H.B. Talbot) Manjón, G. Moreno & Hjortstam and *H. tropicum* Z.Y. Duan & C.L. Zhao are similar to *H. niveomarginatum* by having the moniliform cystidia. However, *H. litschaueri* differs in its larger moniliform cystidia (60–100 × 6–8 µm) and larger subcylindrical basidiospores (9–12 µm × 3–4 µm) [36]; *H. malenconii* is separated from *H. niveomarginatum* by having the dendrohyphidia and larger subcylindrical basidiospores (12–15 × 5.5–10 µm) [37]; *H. membranaceum* is distinct from *H. niveomarginatum* by tuberculate hymenial surface and larger basidiospores (11–13.5 × 4.5–5.5 µm) [10]; *H. moniliforme* differs in its membranous basidiomata and having the larger moniliform cystidia (85–100 × 6–8 µm) [4]; and *H. tropicum* is distinguished from *H. niveomarginatum* by having tuberculate hymenial surface and larger basidia (29.5–38 × 4–6 µm) [13].

*Hyphoderma floccosum* C.L. Zhao & Q.X. Guan, *H. mopanshanense* C.L. Zhao, *H. sinense* and *H. transiens* are similar to *H. niveomarginatum* by having the ceraceous basidiomata. However, *H. floccosum* differs in *H. niveomarginatum* by farinaceous hymenial surface and two types of cystidia: septate cystidia and tubular cystidia [11]; *H. mopanshanense* is separated from *H. niveomarginatum* by porulose hymenial surface and wider thick-walled generative hyphae (4–6 µm) [9]; *H. sinense* is distinct from *H. niveomarginatum* by having two types of cystidia: encrusted cystidia and moniliform cystidia, and cylindrical to allantoid basidiospores [11]; and *H. transiens* is distinguished from *H. niveomarginatum* by having odontioid hymenophore, larger cystidia (40–95 × 7–9 µm) and cylindrical basidiospores (8.5–12 × 2.7–4.1) [38].

***Hyphoderma sordidum*** Y. Yang & C.L. Zhao, sp. nov. Figure 4 and Figure 5.

MycoBank no.: 849949.

**Holotype—**China, Yunnan Province, Qujing, Zhanyi District, Yanzhu Village. GPS coordinates: 25°44′ N, 103°36′ E; altitude: 1950 m asl. On fallen unidentified angiosperm branch, leg. C.L. Zhao, 7 March 2023, CLZhao 27390 (SWFC).

**Etymology—*sordidum*** (Lat.): referring to the sordid white hymenial surface.

**Basidiomata—**Annual, resupinate, adnate, membranous, odorless and up to 5 cm long, 2.5 cm wide and 50–150 µm thick. Hymenial surface smooth, white to cream when fresh, cream upon drying. Sterile margin white, fimbriate, up to 1–2 mm wide.

**Hyphal system—**Monomitic; generative hyphae with clamp connections; colorless, thin-walled, interwoven, 1.5–2 µm in diameter; IKI–, CB–, tissues unchanged in KOH.

**Hymenium—**Cystidia tubular, basally widened, tapering but without sublate apex, slightly sinuous, colorless, thin-walled, 42–72.5 × 6–11 µm; basidia clavate to subcylindrical, constricted in the middle to somewhat sinuous, with four sterigmata and a basal clamp connection, 8–14 × 3–3.5 µm; basidioles in shape similar to basidia, but slightly smaller.

**Spores—**Basidiospores ellipsoid, colorless, thin-walled, smooth, some with irregular oil droplets inside, IKI–, CB–, (2.5–)3–4.5 × (1.5–)2–3(–4) µm, L = 3.62 µm, W = 2.35 µm, Q = 1.56–1.70 (n = 60/2).

**Additional specimens examined (paratypes)—**China, Yunnan Province, Qujing, Zhanyi District, Yanzhu Village. GPS coordinates: 25°44′ N, 103°36′ E; altitude: 1950 m asl. On fallen unidentified angiosperm branches, leg. C.L. Zhao, 7 March 2023, CLZhao 17908; CLZhao 27379 (SWFC).

**Notes—***Hyphoderma anthracophilum* (Bourdot) Jülich, *H. cremeoalbum* (Höhn. & Litsch.) Jülich, *H. multicystidium* (Hjortstam & Ryvarden) Hjortstam & Tellería, *H. tropicum* and *H. obtusiforme* J. Erikss. & Å. Strid are similar to *H. sordidum* by having ellipsoid basidiospores. However, *H. anthracophilum* differs from *H. sordidum* by the cracked hymenophore and larger basidiospores of 6–9 × 4–6 µm [39]; *H. cremeoalbum* is distinct from *H. sordidum* by having larger basidia (30–45 × 7–9 µm) and larger basidiospores (10–14 × 5–6 µm) [5]; *H. multicystidium* is separated from *H. sordidum* by the reticulate and tomentose hymenial surface, wider generative hyphae (2–3 µm) and larger basidiospores (8–10 × 4.5–5 µm) [40]; *H. obtusiforme* differs in having wider generative hyphae (3–4 µm), larger basidia (30–40 × 6–8 µm) and larger basidiospores (10–12 × 5–6 µm) [38]; and *H. tropicum* is distinguished from *H. sordidum* by having tuberculate hymenial surface, moniliform cystidia and larger basidiospores (6.5–7.5 × 3–4 µm) [13].

*Hyphoderma crystallinum*, *H. marginatum*, *H. membranaceum, H. moniliforme* and *H. tenuissimum* C.L. Zhao & Q.X. Guan are similar to *H. sordidum* by having membranous basidiomata. However, *H. crystallinum* differs in *H. sordidum* by hymenial surface with scattered nubby crystals, larger basidia (21.5–31 × 6–8.5 µm) and larger allantoid basidiospores (11–14.5 × 4–5.5 µm) [10]; *H. marginatum* is separated from *H. sordidum* by having cracking hymenial surface, cylindrical cheilocystidia and larger basidiospores (9–10 × 3.5–4.5 µm) [13]; *H. membranaceum* is distinct from *H. sordidum* by having cracking hymenial surface, moniliform cystidia and larger basidiospores (11–13.5 × 4.5–5.5 µm) [10]; *H. moniliforme* differs from *H. sordidum* by having cracking hymenial surface, larger basidia (20–30.5 × 6–7.5 µm) and larger basidiospores (6–9 × 3–4.5 µm) [4]; and *H. tenuissimum* is distinguished from *H. sordidum* by having tuberculate to minutely grandinioid hymenial surface, larger cylindrical cystidia (50–220 × 6.5–13 µm) and larger cylindrical basidiospores (7–10.5 × 3–4.5 µm) [12].

***Hyphoderma weishanense*** Y. Yang & C.L. Zhao, sp. nov. Figure 6 and Figure 7.

MycoBank no.: 849950.

**Holotype—**China, Yunnan Province, Dali, Weishan County, Qinghua Town, Green Peacock Reserve. GPS coordinates: 24°52′ N, 100°12′ E; altitude: 1550 m asl. On fallen unidentified angiosperm branch, leg. C.L. Zhao, 18 July 2022, CLZhao 22403 (SWFC).

**Etymology—*weishanense*** (Lat.): refers to the locality (Weishan) of the type specimen.

**Basidiomata—**Annual, resupinate, adnate, membranous when fresh, hard membranous when dry, odorless and up to 11 cm long, 4 cm wide and 50–100 µm thick. Hymenial surface smooth, white when fresh, slightly buff to buff upon drying. Sterile margin thin, white, up to 1–2 mm wide.

**Hyphal system—**Monomitic; generative hyphae with clamp connections; colorless, thin-walled, interwoven, 2.5–3.5 µm in diameter; IKI–, CB–, tissues unchanged in KOH.

**Hymenium—**Cystidia absent; basidia subcylindrical, constricted in the middle to somewhat sinuous, with four short sterigmata and a basal clamp connection, 16.5–18.7 × 4.5–7.5 µm; basidioles in shape similar to basidia, but slightly smaller.

**Spores—**Basidiospores broadly ellipsoid, colorless, thin-walled, smooth, IKI–, CB–, (4–)4.5–8.5(–9) × (3–)4–7(–8) µm, L = 6.25 µm, W = 5.23 µm, Q = 1.20 (n = 30/1).

**Notes—***Hyphoderma floccosum*, *H. obtusiforme*, *H. puerense* C.L. Zhao & Q.X. Guan, *H. transiens* and *H. tropicum* are similar to *H. weishanense* by having the ellipsoid basidiospores. However, *H. floccosum* differs in *H. weishanense* by having ceraceous basidiomata, farinaceous hymenial surface and two types of cystidia: septate cystidia and tubular cystidia [11]; *H. obtusiforme* is distinguished from *H. weishanense* by having porulose hymenial surface, cylindrical cystidia (50–60 × 8–10 µm), larger basidia (30–40 × 6–8 µm) and larger basidiospores (10–12 × 5–6 µm) [38]; *H. puerense* is separated from *H. weishanense* by the byssoid basidiomata, thick-walled generative hyphae covered by crystals and the tubular cystidia [12]; *H. transiens* is distinct from *H. weishanense* by having the ceraceous basidiomata, odontioid hymenial surface, subcylindrical cystidia and larger basidiospores (9–13 × 3–4.5 µm) [10]; and *H. tropicum* differs from *H. weishanense* by having tubercula hymenial surface, the moniliform cystidia and larger basidia (29.5–38 × 4–6 µm) [13].

*Hyphoderma anthracophilum*, *H. cremeoalbum*, *H. fissuratum, H. sibiricum* (Parmasto) J. Erikss. & Å. Strid and *H. tenuissimum* C.L. Zhao & Q.X. Guan are similar to *H. weishanense* by absent cystidia. However, *H. anthracophilum* is separated from *H. weishanense* by the pale grey to isabelline hymenial surface and larger basidia (30–40 × 5–7 µm) [5]; *H. cremeoalbum* differs in *H. weishanense* by having larger basidia (30–45 × 7–9 µm) and larger basidiospores (10–14 × 5–6 µm) [5]; *H. fissuratum* is distinct from *H. weishanense* by having ceraceous basidiomata, larger basidia (24–28 × 4–4.5 µm) and cylindrical basidiospores [12]; and *H. sibiricum* is distinguished from *H. sordidum* by the small irregular patches of basidiomata and larger basidia (25–35 × 5–7 µm) [38].

## 4. Discussion

Based on the molecular systematics study amplifying the nLSU, ITS and RPBl genes, the family-level classification for the order Polyporales (Basidiomycota) revealed that the four taxa of *Hyphoderma macaronesicum*, *H. medioburiense*, *H. mutatum* (Peck) Donk and *H. setigerum* nested into the family Hyphodermataceae within the residual polyporoid clade [18]. In the present study, from the phylogram inferred from the ITS+nLSU+mt-SSU+RPB1+RPB2 data, three new species grouped into *Hyphoderma* (Figure 1), in which *H. niveomarginatum* grouped with two taxa, *H. membranaceum* and *H. sinense*, and then closely grouped with *H. transiens*, *H. amoenum* and *H. fissuratum*; *H. sordidum* clustered with *H. nudicephalum*; and *H. weishanense* grouped with *H. crystallinum*, and then grouped closely with *H. variolosum*, *H. marginatum*, *H. medioburiense*, *H. assimile* and *H. subsetigerum*. However, morphologically, *H. membranaceum* is distinct from *H. niveomarginatum* by membranous basidiomata and wider basidiospores (11–13.5 × 4.5–5.5 µm) [10]; *H. sinense* differs from *H. niveomarginatum* by membranous basidiomata and thick-walled generative hyphae [11]; *H. transiens* is separated from *H. niveomarginatum* by its odontioid hymenial surface and larger subcylindrical cystidia (50–70 × 6–8 µm) [36]; *H. amoenum* is separated from *H. niveomarginatum* by membranous basidiomata and larger basidiospores measuring 12–15 × 4–6 µm [41]; and *H. fissuratum* differs from *H. niveomarginatum* by leathery basidiomata and cylindrical basidiospores [9]. *Hyphoderma nudicephalum* is distinct from *H. sordidum* by having the farinaceous to odontioid hymenial surface, and conspicuous capitate cystidia with the nonincrusted apices and thick-walled generative hyphae [42]. *Hyphoderma crystallinum* differs from *H. weishanense* by the hymenial surface with scattered nubby crystals and larger, allantoid basidiospores (11–14.5 × 4–5.5 µm) [10]; *H. variolosum* is separated from *H. weishanense* by its tuberculiform hymenial surface and having the tubular cystidia [43]; *H. marginatum* differs from *H. weishanense* by the cream hymenial surface and the larger basidiospores of 9–10 × 3.5–4.5 µm [13]; *H. medioburiense* is distinguished from *H. weishanense* by the porulose hymenial surface and having the tubular cystidia [44]; *H. assimile* is distinct from *H. weishanense* by its white to pale cream basidiomata, the tubular cystidia and larger basidia (33–37 × 7–8 µm) [45]; and *H. subsetigerum* differs from *H. weishanense* by its grandinioid hymenophore with the whitish to ivory yellow hymenial surface, and the thick-walled generative hyphae [37].

Wood-inhabiting fungi are found in living trees, decorticated wood of dead tree branches and trunks as well as manufactured wood products. These fungi of the cell walls and the components within the living cells secrete various enzymes that effectively degrade cellulose, hemicellulose and lignin into simple inorganic substances, and consequently play an important role in forest ecosystems as an important group of decomposers [46]. In terms of geographical distribution and ecological importance, *Hyphoderma* species are an extensively studied group, mainly found on hardwood, although a few species grow on coniferous wood [10]. We believe that more species of *Hyphoderma* occur in subtropical and tropical Asia, since wood-inhabiting fungi play a core role in the forest; they are rich in tropical China [47,48,49,50,51,52,53,54,55]; and it is very possible that the same phenomenon exists for *Hyphoderma*.

## Figures and Tables

**Figure 1 jof-09-01044-f001:**
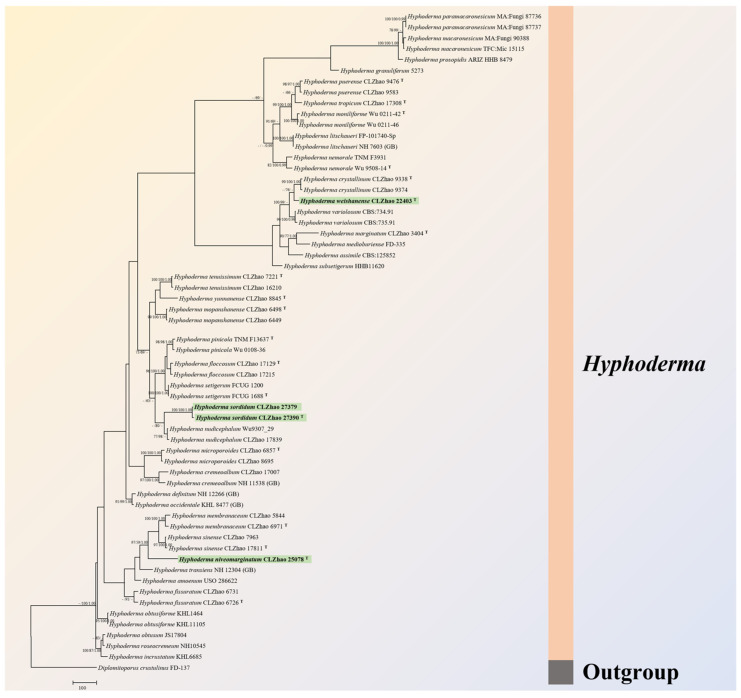
Maximum parsimony strict consensus tree illustrating the phylogeny of three new species and related species in *Hyphoderma* within Polyporales based on ITS+nLSU+mt-SSU+RPB1+RPB2 sequences. The branch is labeled with a maximum likelihood lead value greater than 70%, a reduced lead value greater than 50% and a Bayesian posterior probability greater than 0.95. The new species are in bold/green, the holotypes superscript “T”.

**Figure 2 jof-09-01044-f002:**
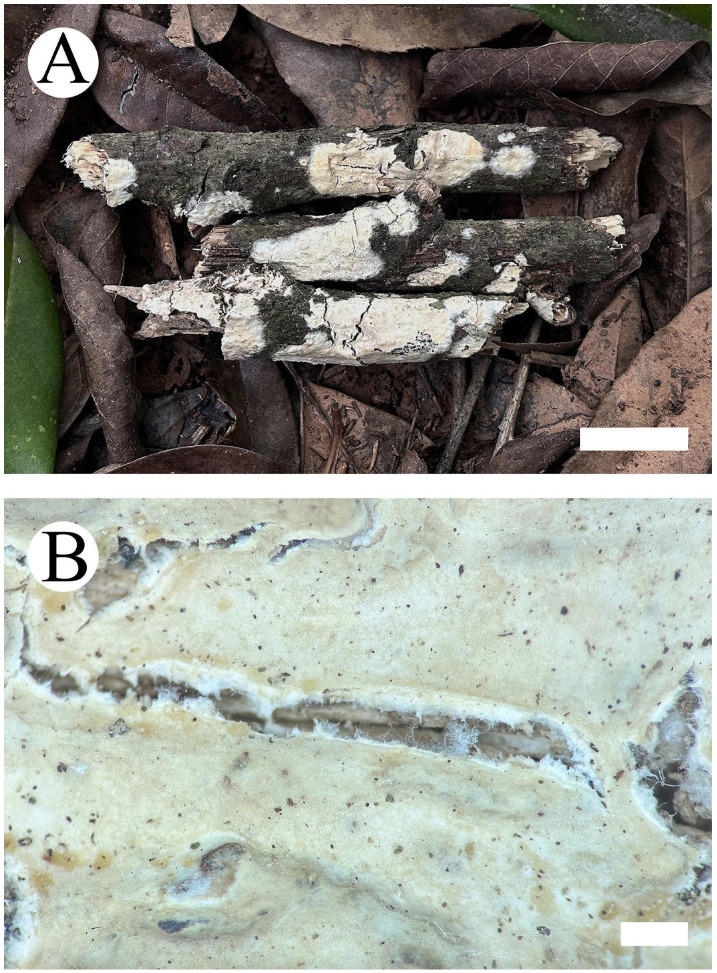
*Hyphoderma niveomarginatum* (holotype): basidiomata on the substrate (**A**), macroscopic characteristics of hymenophore (**B**). Bars: (**A**) = 2 cm and (**B**) = 1 mm.

**Figure 3 jof-09-01044-f003:**
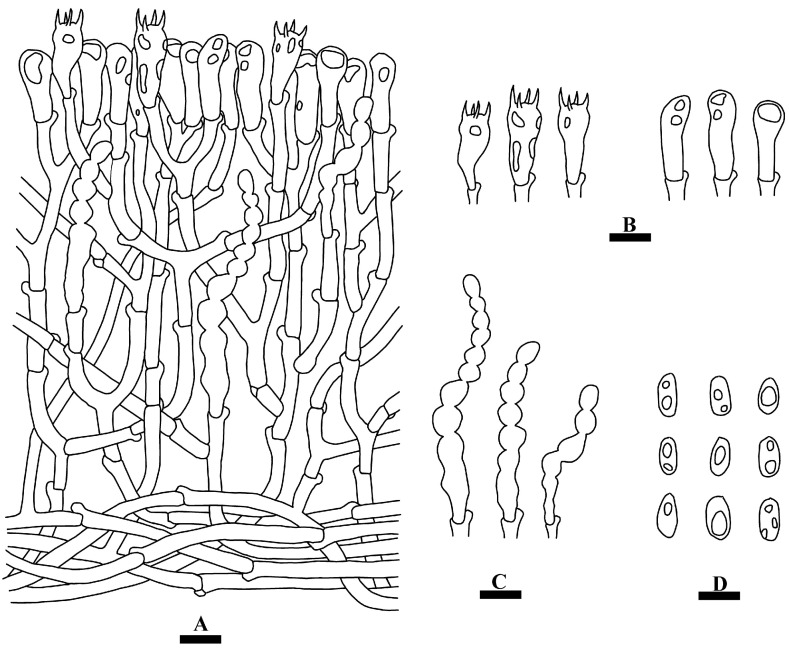
Microscopic structures of *Hyphoderma niveomarginatum* (holotype): a section of the hymenium (**A**), basidia and basidioles (**B**), cystidia (**C**), basidiospores (**D**). Bars: (**A**–**D**) = 10 µm.

**Figure 4 jof-09-01044-f004:**
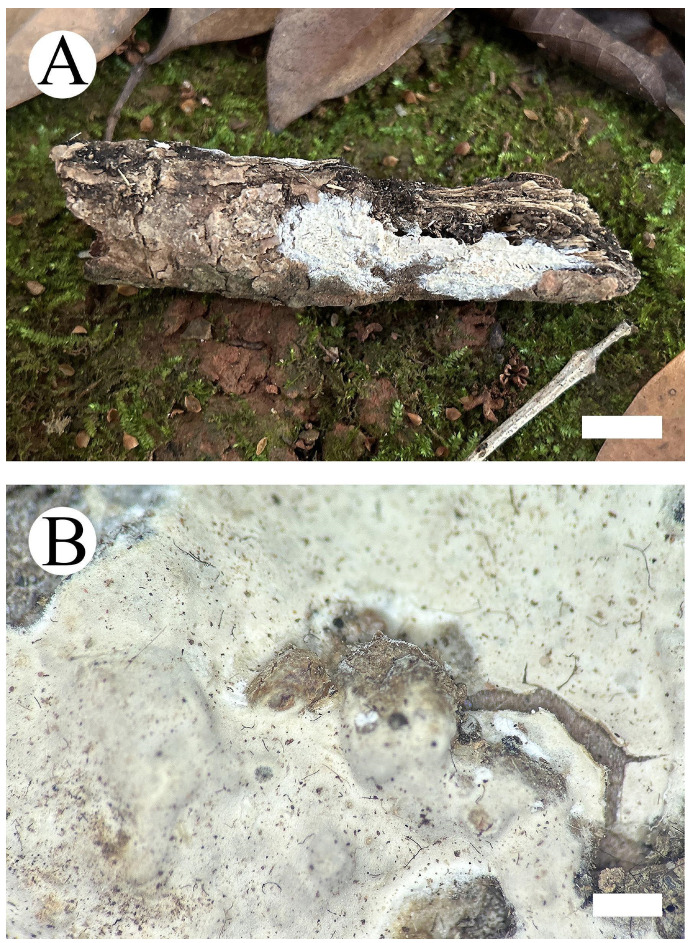
*Hyphoderma sordidum* (holotype): basidiomata on the substrate (**A**), microscopic characteristics of hymenophore (**B**). Bars: (**A**) = 1 cm and (**B**) = 1 mm.

**Figure 5 jof-09-01044-f005:**
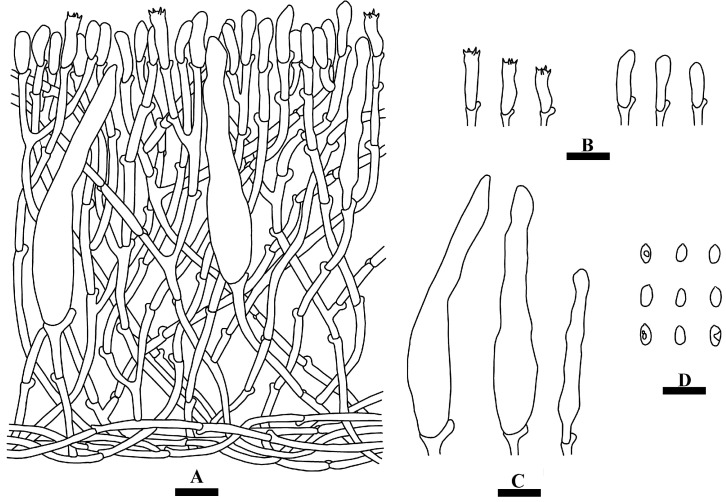
Microscopic structures of *Hyphoderma sordidum* (holotype): a section of the hymenium (**A**), basidia and basidioles (**B**), cystidia (**C**), basidiospores (**D**). Bars: (**A**–**D**) = 10 µm.

**Figure 6 jof-09-01044-f006:**
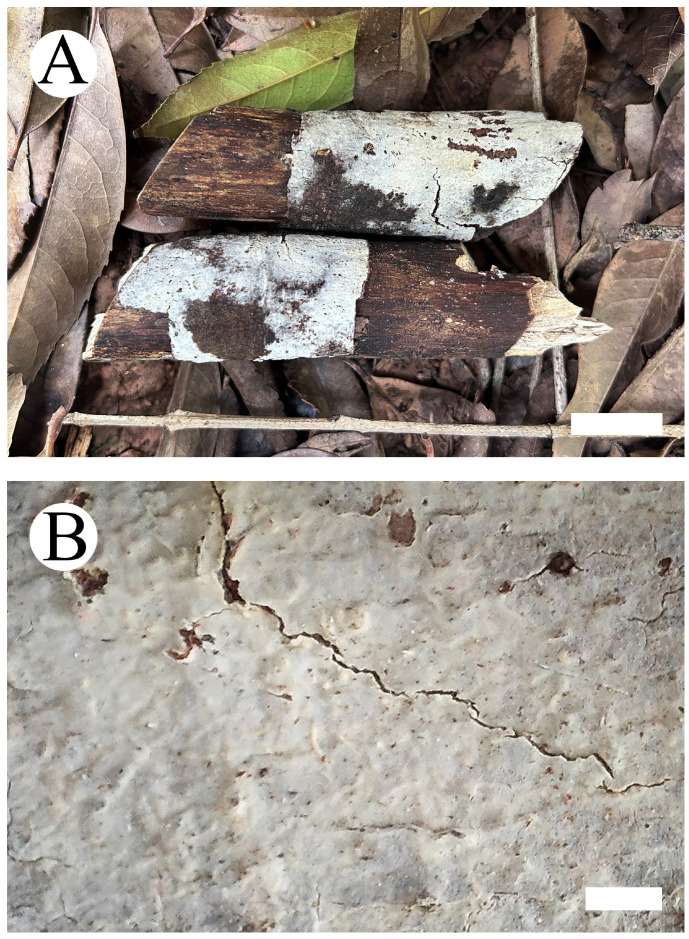
*Hyphoderma weishanense* (holotype): basidiomata on the substrate (**A**), microscopic characteristics of hymenophore (**B**). Bars: (**A**) = 2 cm and (**B**) = 1 mm.

**Figure 7 jof-09-01044-f007:**
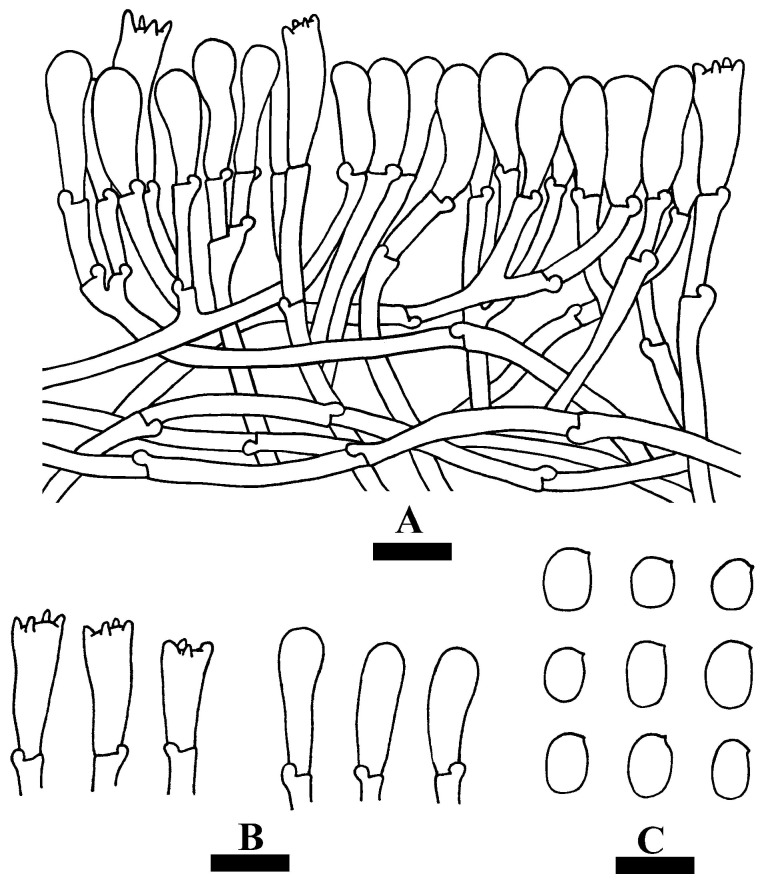
Microscopic structures of *Hyphoderma weishanense* (holotype): a section of the hymenium (**A**), basidia and basidioles (**B**), basidiospores (**C**). Bars: (**A**–**C**) = 10 µm.

**Table 1 jof-09-01044-t001:** List of species, specimens and GenBank accession numbers of sequences used in this study. The new species are in bold, the holotypes superscript “T”.

Species Name	Sample No.	GenBank Accession No.					References
		ITS	nLSU	RPB1	RPB2	mt-SSU	
*Diplomitoporus crustulinus*	FD-137	KP135299	KP135211	KP134883			[18]
*Hyphoderma amoenum*	USO 286622	HE577030					[16]
*H. assimile*	CBS:125852	MH863808	MH875272				[25]
*H. cremeoalbum*	NH 11538 (GB)	DQ677492	DQ677492				[15]
*H. cremeoalbum*	CLZhao 17007	OM985716	OM985753			OQ706819	[13]
*H. crystallinum*	CLZhao 9338 ^T^	MW917161	MW913414				[10]
*H. crystallinum*	CLZhao 9374	MW917162	MW913415				[10]
*H. definitum*	NH 12266 (GB)	DQ677493	DQ677493				[15]
*H. fissuratum*	CLZhao 6731	MT791331	MT791335			OQ706806	[9]
*H. fissuratum*	CLZhao 6726 ^T^	MT791330	MT791334			OQ706805	[9]
*H. floccosum*	CLZhao 17129 ^T^	MW301683	MW293733			OQ706826	[11]
*H. floccosum*	CLZhao 17215	MW301687	MW293735			OQ706829	[11]
*H. granuliferum*	5273	JN710545	JN710545			JN710673	[17]
*H. incrustatum*	KHL6685		AY586668				[17]
*H. litschaueri*	NH 7603 (GB)	DQ677496	DQ677496				[15]
*H. litschaueri*	FP-101740-Sp	KP135295	KP135219	KP134868	KP134965		[13]
*H. macaronesicum*	MA:Fungi 90388	KC984327	KF150025		KF181122		Unpublished
*H. macaronesicum*	TFC:Mic 15115	HE577011	KF150050		KF181118		[17]
*H. marginatum*	CLZhao 3404 ^T^	OM985717	OM985754				[13]
*H. medioburiense*	FD-335	KP135298	KP135220	KP134869	KP134966		[26]
*H. membranaceum*	CLZhao 5844	MW917167	MW913420			OQ706797	[10]
*H. membranaceum*	CLZhao 6971 ^T^	MW917168	MW913421			OQ706807	[10]
*H. microporoides*	CLZhao 6857 ^T^	MW917169	MW913422				[10]
*H. microporoides*	CLZhao 8695	MW917170	MW913423				[10]
*H. moniliforme*	Wu 0211-42 ^T^	KC928282					[4]
*H. moniliforme*	Wu 0211-46	KC928284	KC928285				[4]
*H. mopanshanense*	CLZhao 6498 ^T^	MT791329	MT791333				[9]
*H. mopanshanense*	CLZhao 6449	OM985720	OM985759			OQ706803	[13]
*H. nemorale*	TNM F3931	KJ885183	KJ885184				[4]
*H. nemorale*	Wu 9508-14 ^T^	KC928280	KC928281				[4]
** *H. niveomarginatum* **	**CLZhao 25078 ^T^**	**OR141728**	**OR506179**		**OR543992**		**Present study**
*H. nudicephalum*	Wu9307_29	AJ534269					[27]
*H. nudicephalum*	CLZhao 17839	OM985721	OM985760			OQ706835	[13]
*H. obtusiforme*	KHL1464	JN572909					[17]
*H. obtusiforme*	KHL11105	JN572910					[17]
*H. obtusum*	JS17804		AY586670				[17]
*H. occidentale*	KHL 8477 (GB)	DQ677499	DQ677499				[15]
*H. paramacaronesicum*	MA:Fungi 87736	KC984399					[8]
*H. paramacaronesicum*	MA:Fungi 87737	KC984405					[8]
*H. pinicola*	TNM F13637 ^T^	KJ885181	KJ885182				[17]
*H. pinicola*	Wu 0108-36	KC928278	KC928279				[17]
*H. prosopidis*	ARIZ HHB 8479	HE577029					[4]
*H. puerense*	CLZhao 9476 ^T^	MW443045					[12]
*H. puerense*	CLZhao 9583	MW443046	MW443051				[12]
*H. roseocremeum*	NH10545		AY586672				[17]
*H. setigerum*	FCUG 1200	AJ534273					[27]
*H. setigerum*	FCUG 1688 ^T^	AJ534272					[27]
*H. sinense*	CLZhao 7963	MW301679	MW293730				[11]
*H. sinense*	CLZhao 17811 ^T^	MW301682	MW293732				[11]
** *H. sordidum* **	**CLZhao 27379**	**OR141731**				**OR507165**	**Present study**
** *H. sordidum* **	**CLZhao 27390 ^T^**	**OR141732**	**OR506180**	**OR520149**		**OR507166**	**Present study**
*H. subsetigerum*	HHB11620	GQ409521					[17]
*H. tenuissimum*	CLZhao 7221 ^T^	MW443049	MW443054			OQ706809	[12]
*H. tenuissimum*	CLZhao 16210	MW443050	MW443055				[12]
*H. transiens*	NH 12304 (GB)	DQ677504	DQ677504				[15]
*H. tropicum*	CLZhao 17308 ^T^	OM985727	OM985768				[13]
*H. variolosum*	CBS:734.91	MH862320	MH873992				[25]
*H. variolosum*	CBS:735.91	MH862321	MH873993				[25]
** *H. weishanense* **	**CLZhao 22403 ^T^**	**OR141727**	**OR506181**				**Present study**
*H. yunnanense*	CLZhao 8845 ^T^	OM985769				OQ706811	[13]

## Data Availability

Publicly available datasets were analyzed in this study. This data can be found here: (https://www.ncbi.nlm.nih.gov/; https://www.mycobank.org/page/Simple%20names%20search; http://purl.org/phylo/treebase, submission ID 30751; accessed on 7 September 2023).

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
