# Peer review of "Molecular Systematics and Taxonomic Analyses of Three New Wood-Inhabiting Fungi of Hyphoderma (Hyphodermataceae, Basidiomycota)"

_jof, 2023, doi:10.3390/jof9111044_

Round 1
Reviewer 1 Report
Overall, this manuscript is more-or-less acceptable for publication following some very minor corrections, namely a few words can be changed and a couple sentences can be rewritten for clarity. I have attached a version of the manuscript with my comments and include some below:
- I believe that the ex-type/holotype specimens for other species should also be denoted by a superscript 'T' in the phylogenetic tree (Fig. 1). It is important to know which specimens represent type specimens and are therefore authentic.
- Figure 2 caption: "character ymenophore" has a typo (hymenophore) but besides this "character hymenophore" does not make sense (also see Figure 4, 6 captions). Do you mean "Macroscopic characters of the hymenophore"?
- Figure 6 B: is the white balance a bit off? The hymenophore appears quite yellow but perhaps this is just my monitor.
- I believe that much of the Discussion involving the morphologically comparison of your new species with other related species could also be included in a Notes section in each species description.
- Line 314: This sentence should be rewritten, for example:
"Wood-inhabiting fungi are found in decorticated wood (?) of dead tree branches and trunks as well as manufactured wood products. These fungi secrete various enzymes that effectively degrade cellulose, hemicellulose and lignin into simple inorganic substances, and consequently play an important role in forest ecosystems as an important group of decomposers"

Overall the quality of English language is acceptable but there are a few examples where the sentence could be better written, for example dividing the sentence into two sentences that better convey the different ideas and/or are easier to follow than using several commas. For example, Line 40:
The genus Hyphoderma Wallr. (1833: 576) belongs to the family Hyphodermataceae (Polyporales, Basidiomycota), typified by H. setigerum (Fr.) Donk. (1957: 15), is represented one of the important genera among wood-inhabiting fungi [4].
If you remove the ideas within the commas, it reads:
The genus Hyphoderma Wallr. (1833: 576) belongs to the family Hyphodermataceae (Polyporales, Basidiomycota), is represented one of the important genera among wood-inhabiting fungi [4].
Author Response
|
1. Summary |
|
|
|
||
|
Thank you very much for taking the time to review this manuscript. Please find the detailed responses below and the corresponding revisions/corrections highlighted/in track changes in the re-submitted files.
|
|
||||
|
2. Questions for General Evaluation |
Reviewer’s Evaluation |
Response and Revisions |
|||
|
Does the introduction provide sufficient background and include all relevant references? |
Yes |
Thank you for your evaluation. |
|||
|
Are all the cited references relevant to the research? |
Yes |
Thank you for your evaluation. |
|||
|
Is the research design appropriate? |
Yes |
Thank you for your evaluation. |
|||
|
Are the methods adequately described? |
Yes |
Thank you for your evaluation. |
|||
|
Are the results clearly presented? |
Yes |
Thank you for your evaluation. |
|||
|
Are the conclusions supported by the results? |
Can be improved |
Thank you for your evaluation, and we have revised the conclusions in the revised manuscript. |
|||
- Point-by-point response to Comments and Suggestions for Authors
Comments 1: Page 1, line 14. Revised “3” as “three”.
Response 1: We have revised it according to the reviewer’s comment.
Comments 2: Page 1, line 15. Revised “H. weishanense are proposed” as “H. weishanense, are proposed”.
Response 2: We have revised it.
Comments 3: Page 1, line 17. Revised “the” as “its”.
Response 3: We have revised it according to the reviewer’s comment.
Comments 4: Page 1, line 20. Delete “the” in “presence of the broadly ellipsoid”.
Response 4: We have revised it.
Comments 5: Page 1, line 36. In the sentence “The taxa form Hyphodermataceae”, The taxa from? Or Taxa from the family Hyphodermataceae.
Response 5: Agree, we have revised it as “Taxa from the family Hyphodermataceae”.
Comments 6: Page 1, line 41. Revised “is represented one of the important genera among wood-inhabiting fungi” as “and represented one of the important genera among wood-inhabiting fungi”.
Response 6: We have revised it.
Comments 7: Page 2, line 77. Revised “Fresh fruiting bodies of the fungi growing on the angiosperm branch were collected from the Lincang” as “Fresh fruiting bodies of fungi growing on angiosperm branches were collected from Lincang”.
Response 7: We have revised it.
Comments 8: Page 3, line 102, table 1 and Fig. 1. Indicate which samples are holotypes/ex-types? E.g., with a superscript “T”.
Response 8: We have revised them according to the reviewer’s comments.
Comments 9: Page 6, line 102. Revised “On the fallen angiosperm branch” as “On fallen unidentified angiosperm branch”.
Response 9: We have revised them according to the reviewer’s comment.
Comments 10: Page 7, line 172. Revised “character ymenophore (B)” as “macroscopic characters of hymenophore (B)”.
Response 10: We have revised it.
Comments 11: Page 8, line 212. Revised “On the fallen angiosperm branch” as “On fallen unidentified angiosperm branch”.
Response 11: We have revised it according to the reviewer’s comment.
Comments 12: Page 9, line 219. Revised “character hymenophore (B)” as “macroscopic characters of hymenophore (B)”.
Response 12: We have revised it.
Comments 13: Page 9, line 233. Revised “On fallen angiosperm branch” as “On fallen unidentified angiosperm branch”.
Response 13: We have revised it according to the reviewer’s comment.
Comments 14: Page 11, line 274, Figure 6 (B). Is the white balance a little off in B? It seems a bit too yellowish.
Response 14: We have revised the Figure 6 (B) according to the reviewer’s comment.
Comments 15: Page 11, line 283. Revised “(4.0–)4.5–8.5(–9) × (3.0–)4.0–7.0(–8) µm” as “(4–)4.5–8.5(–9) × (3–)4–7(–8) µm”.
Response 15: We have revised it according to the reviewer’s comment.
Comments 16: Page 12, line 308, Discussion part. I believe that much of the Discussion involving the morphologically comparison of your new species with other related species could also be included in a Notes section in each species description.
Response 16: We have revised it according to the reviewer’s comment, and the morphologically comparison of the new species with other related species are included in a Notes section in each species description.
Comments 17: Page 12, line 309. Revised “Based on the molecular systematics study, amplifying the nLSU, ITS, and RPBl genes, the classification for family-level of the order Polyporales” as “Based on the molecular systematics study amplifying the nLSU, ITS, and RPBl genes, the family-level classification for the order Polyporales”.
Response 17: We have revised it.
Comments 18: Page 12, line 314. Revised “ITS+nLSU+mt-SSU+RPB1+RPB2 data, three new species” as “ITS+nLSU+mt-SSU+RPB1+RPB2 data supported three new species”.
Response 18: We have revised it according to the reviewer’s comment.
Comments 19: Page 13, line 320. Revised “wider basidiospores as 11–13.5 × 4.5–5.5 µm” as “wider basidiospores (11–13.5 × 4.5–5.5 µm)”.
Response 19: We have revised it.
Comments 20: Page 13, line 339. Revised “Wood-inhabiting fungi were generally found in inverted wood and dead tree trunks, artificial wood products, secrete various biological enzymes that degrade cellulose, hemi-cellulose and lignin in wood into simple inorganic substances, and play an important role in forest ecosystems and are important group of ecosystem decomposition” as “Wood-inhabiting fungi are found in decorticated wood of dead tree branches and trunks as well as manufactured wood products. These fungi secrete various enzymes that effectively degrade cellulose, hemicellulose and lignin into simple inorganic substances, and consequently play an important role in forest ecosystems as an important group of decomposers”.
Response 20: We have revised it according to the reviewer’s comment.
Comments 21: Page 13, line 343. Revised “In the geographical distribution” as “In terms of geographical distribution”.
Response 21: We have revised it.
- Response to Comments on the Quality of English Language
Point 1: Overall the quality of English language is acceptable but there are a few examples where the sentence could be better written, for example dividing the sentence into two sentences that better convey the different ideas and/or are easier to follow than using several commas.
Response 1: Thank you for your help, we have revised the manuscript carefully and polish the quality of English language.

Reviewer 2 Report
I did not find significant mistakes and made only some remarks.

Author Response
|
1. Summary |
|
|
|
||
|
Thank you very much for taking the time to review this manuscript. Please find the detailed responses below and the corresponding revisions/corrections highlighted/in track changes in the re-submitted files.
|
|
||||
|
2. Questions for General Evaluation |
Reviewer’s Evaluation |
Response and Revisions |
|||
|
Does the introduction provide sufficient background and include all relevant references? |
Yes |
Thank you for your evaluation. |
|||
|
Are all the cited references relevant to the research? |
Yes |
Thank you for your evaluation. |
|||
|
Is the research design appropriate? |
Yes |
Thank you for your evaluation. |
|||
|
Are the methods adequately described? |
Can be improved |
Thank you for your evaluation, and we have revised the methods part. |
|||
|
Are the results clearly presented? |
Yes |
Thank you for your evaluation. |
|||
|
Are the conclusions supported by the results? |
Yes |
Thank you for your evaluation. |
|||
- Point-by-point response to Comments and Suggestions for Authors
Comments 1: Page 1, line 14. Revised “H. niveomarginatum” as “Hyphoderma niveomarginatum”.
Response 1: We have revised it according to the reviewer's comment.
Comments 2: Page 1, line 55. Revised “Resinicium bicolour” as “Resinicium bicolor”.
Response 2: We have revised it.
Comments 3: Page 2, line 84. How did you the microscopic (microscope type) investigations to obtain the microscopic structures shown in the Figures?
Response 3: We have added more details about the microscopic (microscope type) investigations to obtain the microscopic structures.
Comments 4: Page 6, line 161. Revised “Figure 2 and Figure 3” in “Figures 2 and 3”.
Response 4: We have revised it.
Comments 5: Page 7, line 172. What is ymenophore?
Response 5: We have revised it to “hymenophore”.
Comments 6: Page 12, line 296. Revised “H. sibiricum (Parmasto) J. Erikss. & Å. Strid” as “H. sibiricum (Parmasto) J. Erikss. & Å. Strid”.
Response 6: We have revised it according to the reviewer's comment.
Comments 7: Page 13, line 339. Added “living trees” after “Wood-inhabiting fungi are found in”.
Response 7: We have revised it.
Comments 8: Page 13, line 340. Revised “secrete various biological enzymes that degrade
cellulose” as “These fungi of the cell walls and the components within the living cells secrete various enzymes that effectively degrade cellulose”.
Response 8: We have revised it according to the reviewer's comment.
